# Analogous computations in working memory input, output and motor gating: Electrophysiological and computational modeling evidence

Rachel Rac-Lubashevsky[1,2]*, Michael J. Frank[1,2]

**1** Department of Cognitive, Linguistic & Psychological Sciences, Brown University, Providence, Rhode Island, United States of America, **2** Carney Institute for Brain Science, Brown University, Providence, Rhode Island, United States of America

\* rac.hunrachel@gmail.com

## Abstract

Adaptive cognitive-control involves a hierarchical cortico-striatal gating system that supports selective updating, maintenance, and retrieval of useful cognitive and motor information. Here, we developed a task that independently manipulates selective gating operations into working-memory (input gating), from working-memory (output gating), and of responses (motor gating) and tested the neural dynamics and computational principles that support them. Increases in gating demands, captured by gate switches, were expressed by distinct EEG correlates at each gating level that evolved dynamically in partially overlapping time windows. Further, categorical representations of specific maintained items and of motor responses could be decoded from EEG when the corresponding gate was switching, thereby linking gating operations to prioritization. Finally, gate switching at all levels was related to increases in the motor decision threshold as quantified by the drift diffusion model. Together these results support the notion that cognitive gating operations scaffold on top of mechanisms involved in motor gating.

## Author summary

How do humans decide which information is relevant to attend to in memory, which cognitive operation to take, and when? Flexibly updating, maintenance and retrieval of relevant information from working memory (WM) are thought to be managed by gating computations in the frontostriatal network, supporting higher order learning and cognitive flexibility. Using the reference-back-2 task, we tested the key properties of gating. Namely that they are selective ("content-addressable") and that principles of cognitive "actions" (including input gating of WM, output gating from WM) are scaffold on top of the motor gating operations. Using trial-by-trial EEG indexing and quantitative computational modeling (the hierarchical drift-diffusion model) we showed that action selection at all three levels of gating have separable neural signatures but they operate partly in

**Data Availability Statement:** All behavior and EEG data files used for the analyses of the paper are available from the Dryad, Dataset at: https://datadryad.org/stash/share/

TPUcBEhvZTS8nCGJA9xJoAG7LyfL5Pgo2_ q3A66i-c4; DOI: https://doi.org/10.5061/dryad. 00000002t.

**Funding:** MJF received funding from the National Institutes of Health under award number R01 MH084840-08A1. https://www.nih.gov/grants-funding RRL was supported by THE ISRAEL SCIENCE FOUNDATION (grant No. 96/19) https://www.isf.org.il/#/support-channels/1/10 RRL was supported by Fulbright United states - Israel Educational Foundation. https://fulbright.org.il/program/1/35 The funders had no role in study design, data collection and analysis, decision to publish, or preparation of the manuscript.

**Competing interests:** The authors have declared that no competing interests exist.

parallel, such that decisions about a response are processed to some degree even while the identity of the cognitive rule were uncertain. Furthermore, we showed analogous computations across levels of gating as selection of WM representation and of motor action lead to increase in the estimated decision threshold and to enhanced neural coding of the selected information thereby providing a novel link between WM gating and WM prioritization.

## Introduction

Optimal flexible behavior requires an agent to not only respond to incoming sensory events but to adaptively adjust action selection based on context, including previous events in memory [1]. Moreover, while some events in memory need to be robustly maintained over time in the face of distracting interference, sometimes sensory events dictate that such memories should be disrupted and rapidly updated. This challenge is referred to as the stability vs flexibility tradeoff [2–4] and highlights the need for a context-dependent control mechanism that selectively gates information into and out of working-memory (WM) to guide actions [5–10]. Adaptive control is particularly crucial given a capacity-limited WM system in a complex environment, where only a subset of perceptual information is task-relevant and only a subset of currently maintained WM items may be useful for guiding ongoing behavior.

The PBWM (prefrontal cortex basal ganglia working memory) model is a computational model that leverages powerful mechanisms of dopaminergic reinforcement learning (RL) in basal ganglia (BG) so as to optimize gating policies that control access to and from prefrontal cortex, to be held in WM [9–11]. According to PBWM, gating of both motor and WM actions are implemented by a common canonical set of operations whereby phasic DA signals reinforce BG gating actions that yield successful task performance. A distinguishing characteristic of the PBWM model is that WM gating is an elaboration of the more established BG mechanism of motor gating [7,9,11]. Indeed, according to PBWM, WM gating can be further divided into *input gating*, whereby BG can control access to stored information in PFC, and *output gating*, whereby other BG circuits can control which among several maintained WM items should influence motor decisions (*response gating;* see Fig 1). Moreover, PBWM extensions further assume that frontostriatal gating processes operate across a spectrum of abstract actions ranging from action plans in premotor cortex (PMC) to context-relevant cognitive information (like rules and task sets) in more anterior PFC [11–13]. Various imaging, lesion and pharmacological studies have provided support for PBWM gating mechanisms (see [14–16] for reviews). These cognitive and motor gating circuits are nested within a cortico-striatal hierarchical system that is arranged on a rostral-caudal axis in the frontal lobe (Fig 1) [17–20], such that the posterior response gate is dependent on, and constrained by, the more rostral cognitive gates [6,11,12]. Indeed, anatomical studies have revealed asymmetric topography with more fibers projecting information from anterior PFC to posterior BG [21]. Nevertheless, despite this converging evidence for hierarchical PFC-BG gating interactions, the core assumption that response, input and output gating share computational properties–or how they unfold in time–has not been rigorously tested.

The N-back task is often used for examining variations in executive function [22–23]. While the N-back likely depends on both input and output gating (and can be modeled with PBWM [24]), experimentally, it provides a coarse measurement of WM control processes that are confounded with a variety of complex cognitive processes (e.g., encoding, inhibition, binding, matching, maintenance, updating and removal). To address this issue, and to focus on

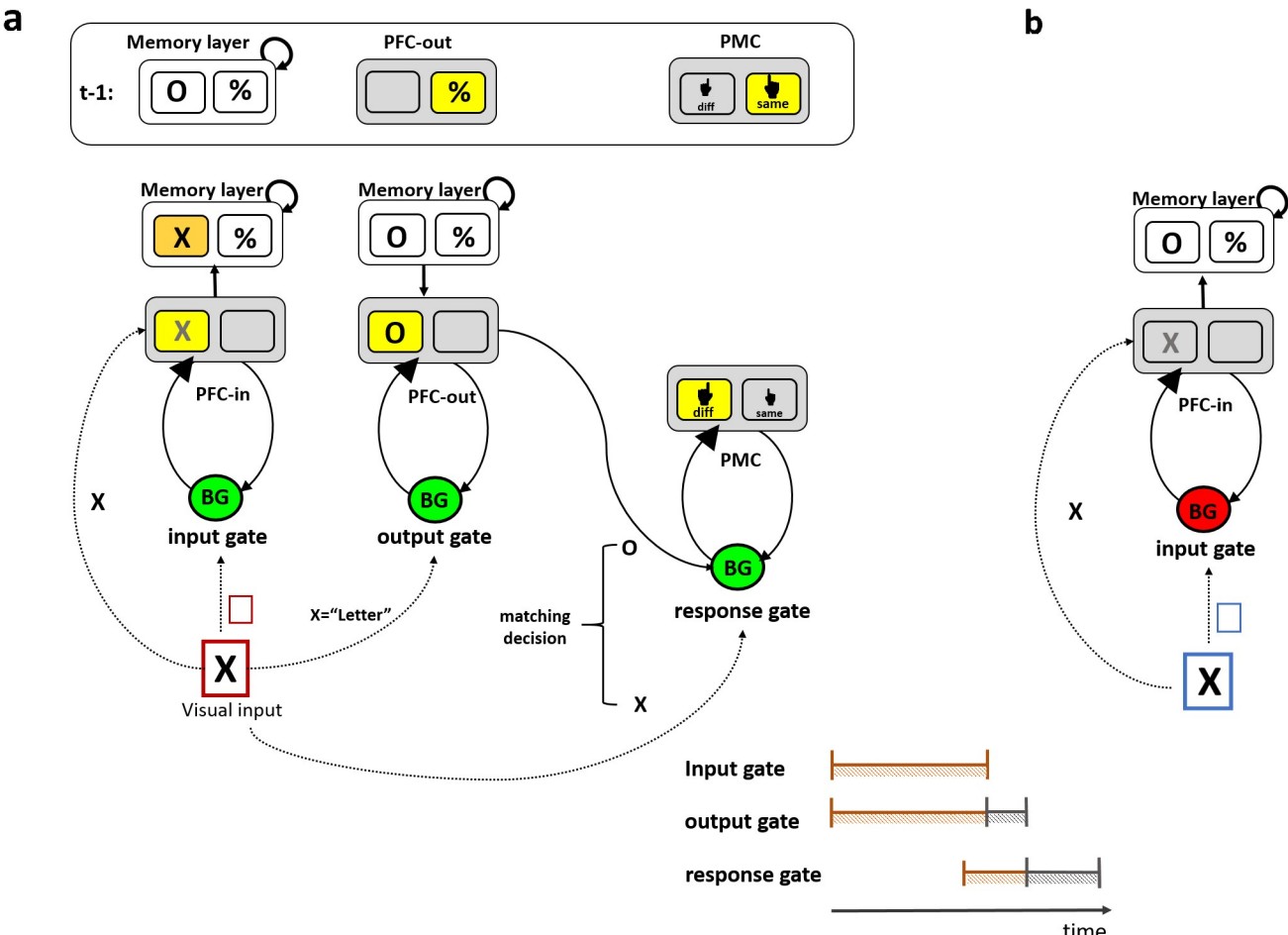

**Fig 1. Schematic of the PBWM architecture for canonical gating operations.** (a) The gating system is composed of three core circuits that are hierarchically arranged from left to right. For illustration the operations needed to solve the task used in this paper are shown. At the beginning of a trial, the memory layer actively maintains the information from the previous trial (e.g., "O" and "%"). Sensory information (e.g., the "X" in the visual input) is transiently represented in a prefrontal cortex layer (labeled "PFC-in"). The input gate controls whether the new visual information (e.g., "X") is updated to be maintained in a PFC memory layer, and if so, to which address or anatomical "stripe" within that layer (selective gating). The colored frame in the visual input represents a task cue signifying the updating policy that should be taken by the input gate. In this example, the red frame indicates that memory should be updated, so the input gate selectively updates the right PFC stripe (corresponding with the letter category, replacing the "O" in memory with "X"). The output gate controls which memory representations is prioritized in PFC-out (e.g., a deep layer of the PFC) to influence subsequent processing (e.g., here the relevant representation in memory is "O" and not "%", given that the input to be compared is in the letter category). The response gate controls which motor response to select (e.g., here "same" (S) or "different" (D)) in posterior prefrontal layer (labeled here PMC), by comparing the output-gated memory representation ("O") with the current visual input ("X"). Gate switching at the output and response gates are reflected by the change in the active stripe between the previous trial (t-1) and in the current trial. (b) An example of a maintenance trial. The blue task cue indicates that the appropriate updating policy is maintenance, and thus sensory information (e.g., "X") is prevented from being updated in PFC memory (the gating signal from the BG is therefore a No-Go, indicated by the red BG). Gate switching in the input gate corresponds with moving between updating state (as shown in a) and maintenance state (as shown in b). Yellow squares indicate active stripe selected by the corresponding gate. The orange square represents the newly updated stripe in memory. The predicted temporal order of the gating loops is depicted at the bottom right of the figure with input and output gating starting roughly at the same time continuing in parallel (indicated in orange) with output gating terminating later (serial termination indicated in grey). The response gate is expected to initiate with some delay due to conflict at upper levels, and to terminate even later than the output gate.

input gating specifically, we recently developed the reference-back task [25–29], a continuous WM updating task. Behavioral, EEG and fMRI data support the notion that the reference-back task taxes cortico-striatal input gating [27,29,30,31]. However, a core distinguishing feature of the PBWM framework from that of other gating models is that it affords *selective* gating, whereby items can be input gated to, and output gated from, distinct addresses in memory

(represented as PFC "stripes" or ensembles in the PBWM model). Here we aimed to augment the reference-back task (see review [32]) to more directly assess these separable content-addressable input and output gating functions, while also assessing how they relate to response gating.

To do so, we amended the reference-back task to mirror the SIR2 task, one of the key tasks developed to illustrate the need for learning selective gating policies in PBWM [10]. To manipulate the need for selective gating in a content-addressable manner, the SIR2 task includes two separate store cues (S1 or S2) in which the associated WM items had to be updated independently to separable WM stores and then accessed independently in response to corresponding recall probes (R1 or R2), in arbitrary order and over intervening distractors [10]. Through RL, PBWM learned a selective input gating policy that placed S1 and S2 items into separate PFC stripes, and an output gating policy that accesses the appropriate stripe given the corresponding R1 or R2 probe, thus implementing content-addressable memory. Given the unpredictable order in which items are stored and recalled, models without selective gating policies cannot solve the SIR2 task. This same selective gating ability was shown to useful to perform complex, hierarchically structured tasks and for supporting generalizability and task structure learning [11,12,33,34]. However, the SIR2 task does not have psychometric properties useful for human experimentation.

The reference-back-2 task (Fig 2) retains all the key features of SIR2 and allows laboratory testing of content-addressable selective input and output gating. The reference-back-2 task, like the original reference-back, is a continuous WM updating task that is composed of two trial types. *Updating trials* require both a matching decision and WM updating, whereas *maintenance trials* are equivalent in their perceptual and decision making demands but do not require WM updating. To introduce the need for selective gating, the reference-back-2 task includes stimuli that belong to two different categories (e.g., symbol/letter), where only one of these representations is relevant on each trial. In updating trials (marked with a red frame), participants need to update only the stimulus in WM belonging to the same category as the presented stimulus, while continuing to maintain that of the other category; thus demanding selective input gating (see Fig 1). The input gate, gates relevant information (e.g., the probed "X") into a specific WM memory address (stripe) given the appropriate context (e.g., the red

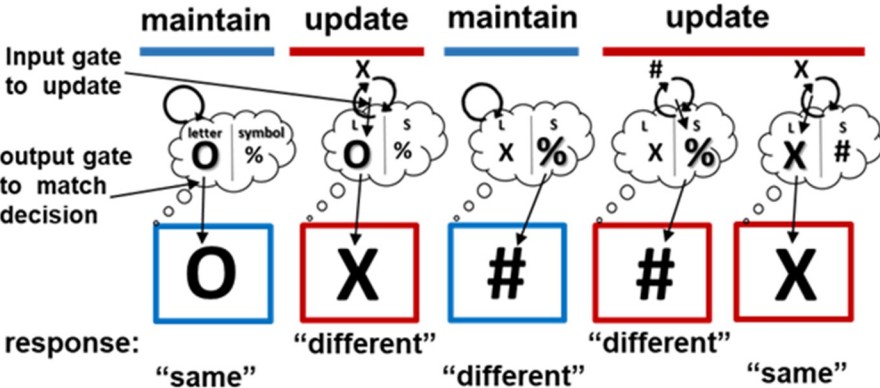

**Fig 2. Trial events in the reference-back-2 task.** Trials with red frame are updating trials, while blue trials are maintenance trials. On each trial, participants make a matching decision between the probe and last stimulus presented in red from the same category. Each side of the thought bubble represents an address (or PFC "stripe") in WM. During updating, input gates assign information to an address based on its category. During output gating, the relevant category is accessed from memory for the decision. The response gate makes the same/different selection. See the method section for more detail.

frame indicates to update). Further, for making the matching decision, only the currently relevant category item should be considered; thus demanding selective output gating. The output gate selects amongst maintained WM representations (in the memory layer) and output gates relevant information for subsequent processing (e.g., given the letter category in the probe, the relevant representation in memory is "O" and not "%"). Finally, for motor responding, the participant needs to compare the probed stimulus ("X") to the WM representation selected by the output gate ("O"), in order to respond "same" or "different", thus demanding selective response gating.

Here, we aim to test four key predictions from PBWM. First, we assessed whether input, output, and response gating share analogous computations that evolve dynamically in time. In particular, in PBWM, gating signals are used to displace prefrontal activity states with new information. Thus, when gating demands (at any level) switch from one trial to the next, a transient period of conflict ensues in which the previous and currently relevant information are competing. As such, PBWM predicts gate switching will be associated with an independent response time cost that arises from PFC conflict analogous to that experienced when needing to switch motor responses in PMC [35]. Indeed, all levels of conflict are thought to recruit the same basal ganglia mechanism that slows down responses and increases decision threshold [12,35]. This prediction is also consistent with findings in which switching between objects in WM during cognitive operations gives rise to a performance cost [36,37]. In that literature, the slowing in switch trials is attributed to "attentional control": the need to move the focus of attention to the other object in WM. Under this view, attention is allocated to only one process at a time thus creating a bottleneck that controls which WM representation to select (i.e., prioritize) and use to guide subsequent cognitive operations or action selection [38,39].

Second, we tested the core PBWM prediction that output gating is used to enhance the representation associated with the currently relevant category (bolded representation in Fig 2). Previous work has shown that focused attention to one WM item prioritizes it. Directing attention to a specific WM content during the retention interval using an external cue (retrocue; see review by [40]) or by instructing participants to refresh ("think of") [41,42] puts such content in a privileged state where it is more resistant to perceptual interference, has better memory strength (reduced errors at retrieval) and heightened accessibility for later use [38,40,43,44]. WM prioritization is marked by sustained, elevated neural activity [45–47] and persistent alpha suppression [48] that increases decodability of the prioritized representations [48–51]. Such an account is consistent with output gating, but to date the gating WM literature and the WM prioritization literature have not been linked. To this end, we aimed to directly test the prediction that output gating (and switches thereof) places demands on prioritization to increase neural discriminability between the two candidate categories. Analogously, we also predicted that response switching will boost the neural indexing of the corresponding motor action representation, given that switches at the motor level place additional demands on response gating.

Third, we tested the prediction that input, output and response gating are processed in parallel. Converging evidence from neural network modeling of hierarchical rule learning [11,12,52], and experimental work with cognitive-control tasks (e.g., [53–55]) suggest that WM and action selection are managed by independent mechanisms, operating mostly in parallel on different levels of information. In biologically inspired models of PBWM [11,12,52], parallel processing can occur when PFC projections to lower level striatum are strong, resulting in faster accumulation of evidence across levels of the hierarchy to favor a single response. The evidence for parallel processing across levels of information is also well documented in task switching literature when task sets and responses both switch [53–55] but also when context and items in WM both switch [56]. Parallel processing in these cases is signified by under-

additive interactions reflecting that the cost of switching two levels of decision was less than the sum of their individual durations.

However, the PBWM modeling simulations and EEG work suggest that processing is not perfectly parallel: indeed, conflict about the task-set in PFC has to be resolved before motor action selection can be informed by the task-set. In the model, the degree of this influence is controlled by projections to the STN leading to an increase in decision threshold that prevents interference in stimulus-response mappings across task-sets [12]. Relatedly, in EEG data, switches in higher level task rules gave rise to anterior EEG signatures which precede those of more posterior motor EEG signatures [35]. Therefore, while there is evidence for simultaneous accumulation of evidence for decision across levels, there is also evidence for serial termination where response decision is delayed by conflict at the higher cognitive level. We aimed to assess whether neural markers of input, output and response gating are separately detectable, and whether their temporal order will also exhibit this partly parallel dynamic.

Finally, we tested whether the behavioral costs associated with gate switching can be assessed more quantitatively by computational models that summarize the impact of BG gating on response time distributions. The behavioral dynamics of BG gating neural networks at the motor level can be approximated by the drift diffusion model (DDM) [57]. Moreover, in these models and related electrophysiological data, premotor conflict (e.g., following switches), lead downstream in the BG to an increase in the effective "decision threshold", buying more time to allow more cautious and deliberate response selection [35,58–60]. Hierarchical PFC-BG models further suggest that this same computation is recruited when conflict occurs at higher level prefrontal task representations, preventing the lower level corticostriatal circuit from selecting actions until such conflict is resolved [12]. Thus according to PBWM, conflict at the WM levels (switches in WM gating) should give rise to an increase in the motor response decision threshold. We thus aimed to test whether neural markers of response switching and conflict are also impacted by switches of WM input and output gating, and whether they are accompanied by a concomitant change in decision threshold.

In sum, the current study was designed to test the hypotheses that selective input, output and response gating are managed by analogous computations and that they are distinguishable temporally. Very little is known on the order in which selective cognitive and action decisions are operating and how they unfold in time. This is important, in part, because input and output gating rely on close cortico-striatal circuits that may require high temporal resolution to disentangle. Furthermore, we investigate the functional role of WM gating and response gating. We employ a trial-by-trial indexing approach of EEG signals to quantify the implications of selective gating on the prioritization of WM and action representations. Finally, we will examine if the same control mechanisms that are engaged to response conflict during motor switching are also recruited in response to cognitive conflict arising from multiple items or task rules held in WM. To quantify these dynamics, we will employ an abstract mathematical model, the hierarchical drift-diffusion model (HDDM) [35,61].

## Results

We first assessed the most basic prediction that switches at each gating level (input, output, response) will translate into separable costs on behavior. Three-way repeated-measures ANOVAs were run on mean RT and mean error rate as a function of switches in input (update, maintain) × output category (letters, symbols) × response (same, different).

For RT, significant main effects were observed for switches at all levels: input ($F_{1,29} = 103.99$, $p < .001$, $\eta_p^2 = .78$), output ($F_{1,29} = 54.06$, $p < .001$, $\eta_p^2 = .65$) and response ($F_{1,29} = 5.40$, $p = .03$, $\eta_p^2 = .16$). The faster RT for repeats than for switches is consistent with the canonical gating

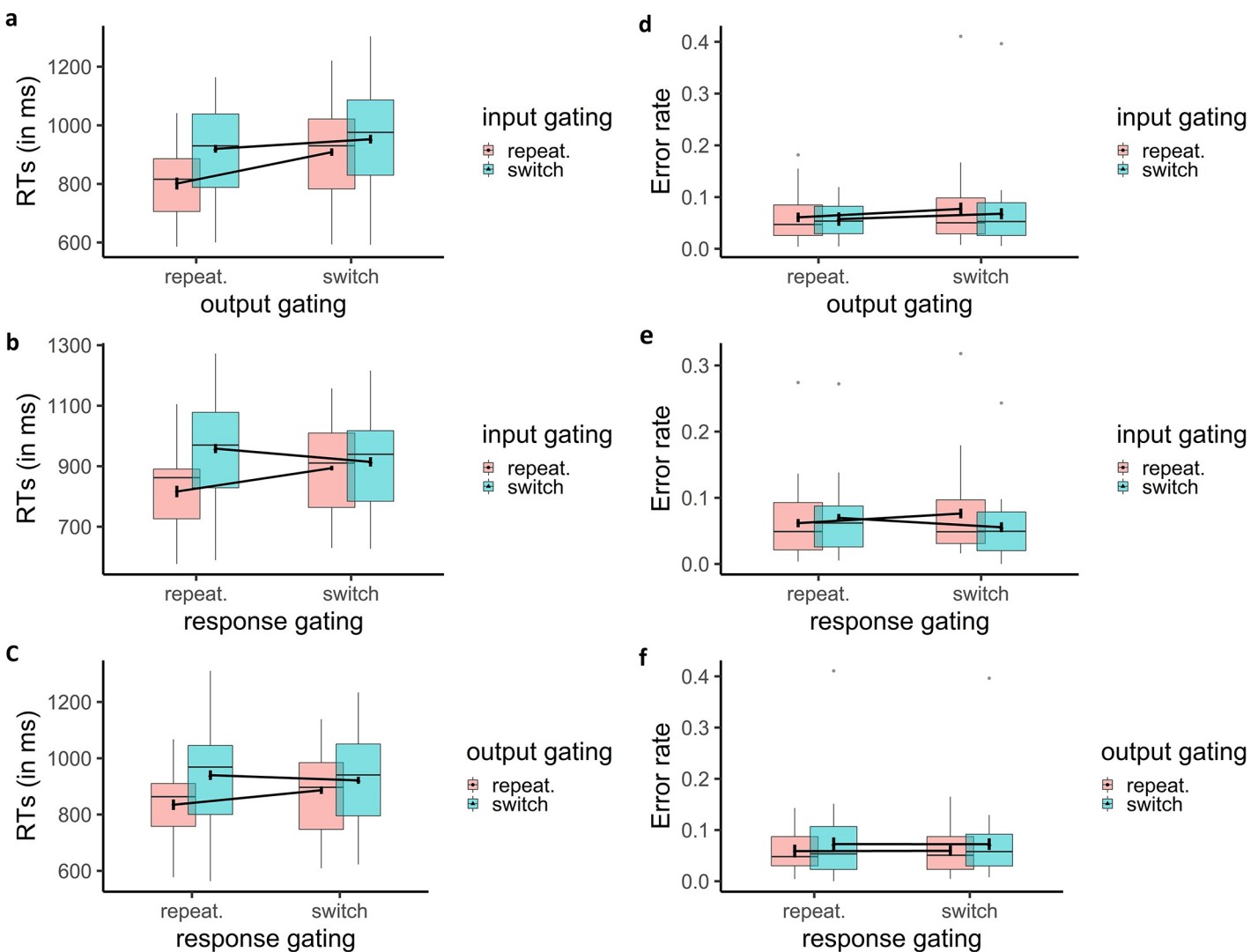

**Fig 3. Behavioral effects of switching in gating demands.** Mean RT (a-c) and error rate (d-f) demonstrate under-additive interactions at all levels (output and response, input and response and, input and output).

model in which a switch at any level incurs conflict and delays motor responding. Significant under-additive interactions (Fig 3) were also observed between input × output ($F_{1,29} = 57.81$, $p < .001$, $\eta_p^2 = .67$), input × response ($F_{1,29} = 81.51$, $p < .001$, $\eta_p^2 = .74$), and output × response ($F_{1,29} = 36.51$, $p < .001$, $\eta_p^2 = .56$). The three-way interaction was also significant ($F_{1,29} = 18.39$, $p < .001$, $\eta_p^2 = .39$) indicating that the under-additive interactions between switches at two levels is limited to the case where the third level was not switching.

For error rate, the only significant effect was for the 2-way input × response interaction ($F_{1,29} = 43.00$, $p < .001$, $\eta_p^2 = .60$), such that switches in input gating demands were related to increased accuracy for response switching (Fig 3). This result indicates a speed-accuracy trade-off given the RT results from that same interaction; this will be further investigated with the DDM model below. None of the main effects nor the other interactions were significant ($F_{1,29} < 3.7$, $p > .06$). Note that an extended five-way ANOVA was also conducted with WM state (updating, maintenance), updating frequency (rare, frequent) and the three gating levels

(switch, repeat) in input, output and response gating. The results of this larger analysis are reported in S1 Text and S1 Fig.

## Temporal evolution of neural gating dynamics across levels

Given that switches in gating demands at all levels had observable effects on behavior, but with under-additive interactions, we also assessed whether such switches would be observable in neural activity. The impact of gating was most observable during switch trials, which induce a transient period of conflict in cortex (the equivalent of an ERP in the model) [12,35,62]. Notably, these models are hierarchical, such that neural signatures of gate switching evolve in time, with higher level gates inducing conflict prior to the response gate (Fig 4A; [62]). As noted in the introduction these processes are predicted to evolve partly in parallel leading to under-additive interaction but also to terminate serially. Indeed, previous behavioral work has showed evidence for a combination of parallel and serial processing among cognitive and response selections, a result that was captured using the same hierarchical PFC-BG neural network model [52].

To evaluate whether gating processes are observable and whether they evolve in parallel or in serial, we focus on trial-to-trial switches that place demands on input, output, and response gating independently. Accordingly, we leverage three orthogonal switch versus repeat trial contrasts to index gating demands: (a) Input gating: The transition from maintenance trials to

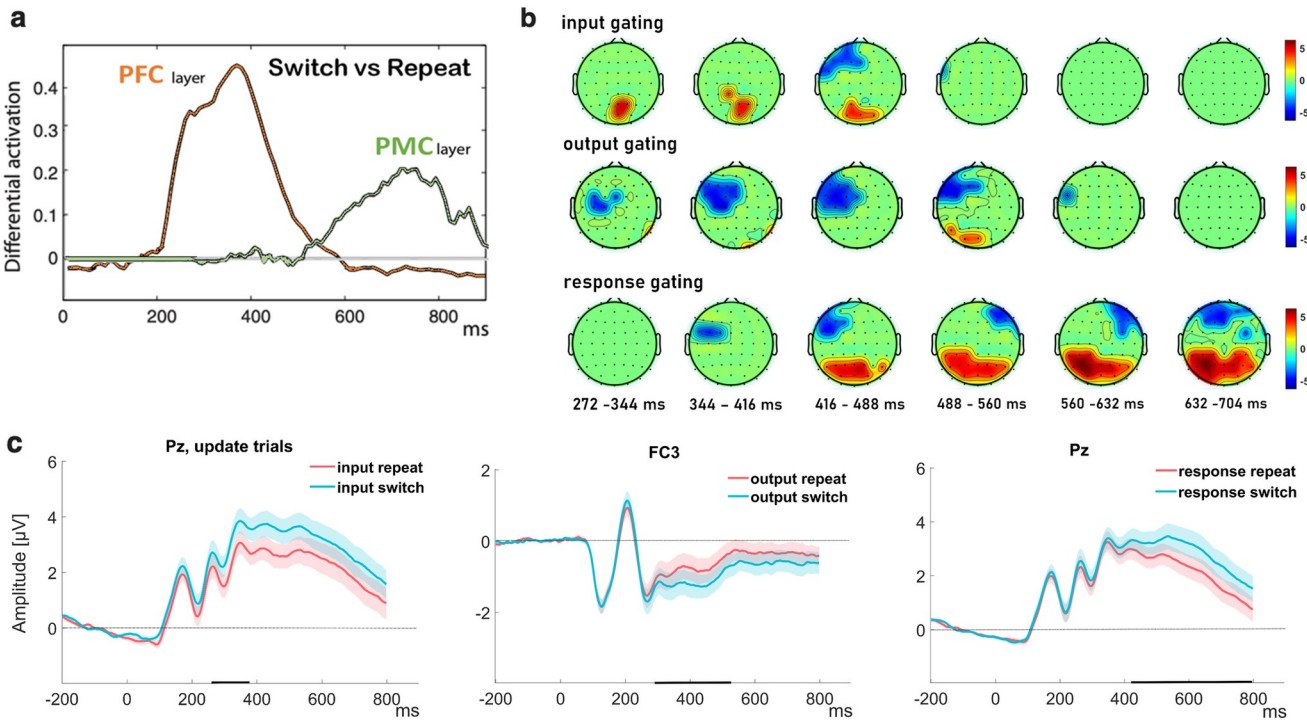

**Fig 4.** (a) Summed activation over units in PFC-BG neural network, showing impact of gate switch vs repeat early in PFC and later in motor cortex (adapted from [62]). (b-c) Current study. (b) Scalp map topography for the predictors that survived permutation correction for the three gating types (input in updating trials, output, response in three rows from top to bottom). The color in the scalp map represent the t-value of the average regression weights in each electrode at the time range indicated above each scalp map. Input, output and response gating signatures evolve sequentially in time, but with clear temporal overlap. (c) Grand ERP plots at representative electrodes from the significant univariate patterns show significant and dissociable effects for switching during input, output and response gating. The shaded error bars represent standard error of the mean (SEM) and the black markers on the x-axis reflect the time points where the difference between switch and repeat was significant.

updating trials requires a shift from a closed to an open input gate, in order to selectively update the task-relevant category. These switch trials are compared to update repeat trials. (b) Output gating: The relevant category for guiding the match decision is selected from within memory using output gating. When the category switches across trials, a new address must be accessed, placing higher demands on the output gate relative to when the category repeats. (c) Response gating: Switches of the same/different response should result in transient conflict at the level of the response decision relative to repeats.

We used a regression approach to extract spatiotemporal clusters that reflect the three gating effects on the trial-by-trial EEG signal, controlling for RT and multiple comparisons (see Methods; [63,64]). The results of this regression analysis yielded significant univariate patterns for WM input gating, output gating and response gating (Fig 4B). Notably, the neural markers of all three gating levels (input, output and response) evolved dynamically in order (Fig 4B), but with substantial overlap in time (410ms– 470ms), consistent with previous empirical studies and hierarchical computational models of cortico-striatal circuitry [12,52]. The response index also persisted on its own after the input and output gate signatures (see Fig 4B). The distinct univariate patterns of activity for each gating level, with largely overlapping time windows, provides neural and temporal support for the PBWM prediction that WM is managed by independent gate selection mechanisms operating mostly in parallel [52]. A closer look at these univariate patterns showed distinct ERP components for each level of gating at sites where the univariate patterns had maximal activity (Fig 4C). Specifically, the switch in the input gate to WM in updating trials showed increase in amplitude in the parietal P3b [65]. The same parietal site also showed a positive effect for response switching but at a later time window in the positive slow wave (PSW) component [66]. Finally, output switching was associated with a frontally negative enhancement in the N2 time range [67]. We speculate about the role of these ERP effects in the Discussion.

## Gating as a prioritization mechanism

Having identified putative neural signatures of gating operations, we next sought to assess the functional impact of gate switching in terms of prioritization of task-relevant representations. The reference-back-2 task employed in this study is suited to study this question: in every trial, two representations are held in WM. Only one, however, is relevant to guide action selection (and to be selectively updated in WM when needed), while the other representation is prospectively relevant and therefore still requires maintenance. This design allowed us to test if selection within WM (output gating) shares the same neural manifestation as action selection (response gating). We leveraged a trial-by-trial indexing approach [63,64]. Thus in contrast to the last section, in which we identified neural signatures of gate switching, here we used GLM to extract spatiotemporal clusters in the EEG signal that reflect the representations themselves (e.g., relevant category and response), controlling for RT, WM state (update, maintenance) and multiple comparisons ([63,64] see methods).

Supporting the notion that switches at the relevant level enhance the need for gating, we found that the trial-wise neural similarity increased following switches of the relevant gate (see Fig 5B and 5C). Specifically, output switching enhanced the neural difference between the two categories whereas response switching enhanced the neural difference between the two motor action representations. These increases in discriminability were very transient in maintenance trials (only around 200ms) while in updating trials, the benefit for output switching lasted throughout the trial (180-450ms; see Fig 5B). This finding is consistent with the PBWM model scheme in which selective updating of WM further enhances activity in the memory layer (Fig 1) which is compounded with impact of output and response switching operations.

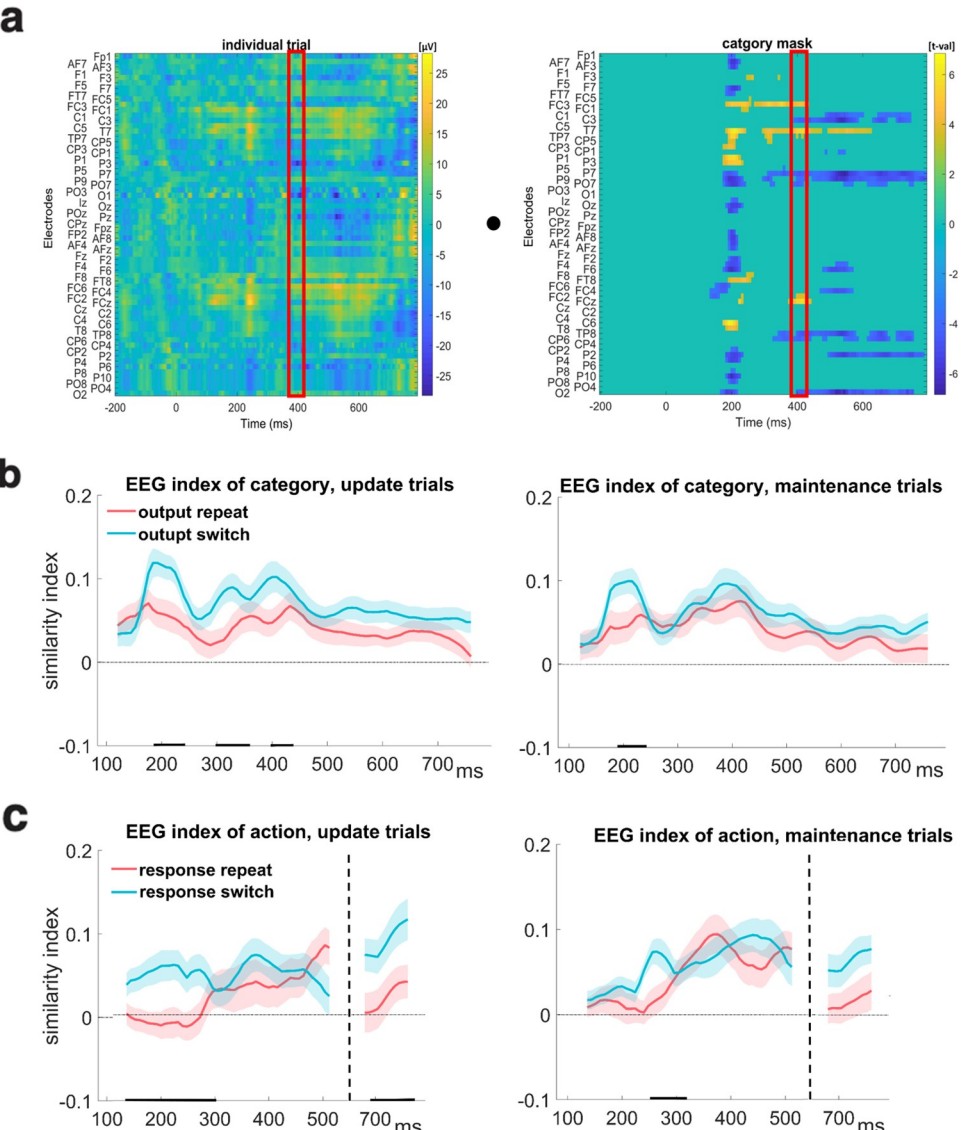

**Fig 5. Temporal dynamics of neural prioritization by gating.** (a) Trial-by-trial indexing of gated representations. Univariate patterns were identified for each representation of interest using GLM. Trial by trial similarity indices obtained by computing the dot-product between the 2D voltage-time map on individual trials with that of the mask for the relevant representation. As an example, the similarity computation with the category mask is shown. This similarity index is obtained over sliding window time bins (red rectangle). The result is a temporally evolving EEG index of similarity in time points for which the mask was significant (see methods for more detail). (b-c) The temporal dynamics of similarity indices to the relevant representations in updating (left) and maintenance (right) trials. (b) The neural similarity to the selected category enhanced following switches in the output gate. (c) The neural similarity to the selected action enhanced following switches in the response gate. Note that the GLM mask of action was significant in two time windows (in an early window between 100-500ms and in a later window 670-800ms) that are separated by the black dashed line (in between these periods the mask was not significant, and hence there is no similarity to plot). The shaded error bars represent standard error of the mean (SEM). The black markers on the x-axis reflect the time points where the similarity index differed between switch and repeat (see methods for more detail).

Together these results support the notion that selective gating at both cognitive and the motor level share the functional role of prioritization that increases the neural index of the representation selected by the relevant gate.

## Gate switching at any level increases decision threshold

The behavioral results described above revealed that switches at any level of gating were related to increased RT cost. To further decompose separable cognitive processes that give rise to such changes in RT, we leveraged the drift diffusion model (DDM). As noted in the introduction, various studies have suggested that dorsomedial frontal cortex detects response conflict (which is elevated following switches) and it recruits prefrontal sub-thalamic pathway mechanisms that increase the effective decision threshold [57–59]. In the DDM, changes in the threshold are used as a cognitive control mechanism that regulates the speed-accuracy tradeoff, with larger thresholds related to slower but more accurate decisions. Importantly this threshold parameter is separable from impacts of other parameters (such as the rate of evidence accumulation or "drift rate", whereby a slower drift rate would lead to slower but less accurate decisions). Thus to properly test out PBWM prediction, we relied on the joint distribution of RTs and error rates and fit the DDM.

To illustrate the functional importance of the decision threshold on optimal behavior during conflict, consider the following example. Imagine you are driving in the morning from home on your routine way to work but this morning you have to first stop by the dentist for a scheduled appointment. You get to the intersection near your work place where you always take a left to get to work so the visual information of the familiar intersection gives rise to a prepotent response to turn left. However, now you also need to consider your goal in WM that you first need to get to your dentist appointment. You then need to retrieve the directions to the dentist from memory before you can decide if it requires going left or right. This situation is referred to as a response conflict because the incorrect habitual response to turn left competes with the context-dependent correct response, which may be to turn right. Such conflict is thought to be managed by an interaction between the medial PFC and the STN that can temporarily pause the response gate by sending diffused excitatory projections to the GPi through a "hyperdirect" pathway that bypasses the striatum. This excitatory projection to the GPi, temporarily suppresses the response gate [68,69] preventing it from selecting any response including the premature response that was activated based on the visual input alone. Neural models and previous empirical findings demonstrated that such response conflict is reflected by a mid-frontal EEG signature that triggers the increase of the decision threshold on the response gate via the mPFC-STN network [35,58,70–74]. The additional time provided by the increased decision threshold, allows a more cautious and deliberate response selection that also considers the evidence provided by WM information. An important extension of previous findings that we aimed to test here was whether information selection to or in WM by the input and the output gate, also manifests as a cognitive conflict that delays responses until the WM conflict is resolved, as predicted by hierarchical models of PFC-BG [12].

Bayesian parameter estimation with HDDM revealed that the DDM provided an adequate fit to choice proportions and response time distributions (Figs 6 and S2 and S3). Moreover, switches at each independent level of gating (input, output, and response) were related to increased decision thresholds (see Fig 7 for statistics). Notably, in updating trials, such effects were incrementally larger when switches were closer to the response level. This result supports the notion that response conflict has preferential impact on motor decision threshold, but that switches in higher level gates can nevertheless recruit the same process.

Relatedly, parameter estimates revealed under-additive interactions (Fig 7) that mirrored those described in behavioral summary statistics above, and are also consistent with the EEG findings that gate switches overlapped in time. Specifically switching both input and output gates increased threshold to a similar level as switching only one gate, again suggesting that input and output gating in WM are two independent selection mechanisms that can be

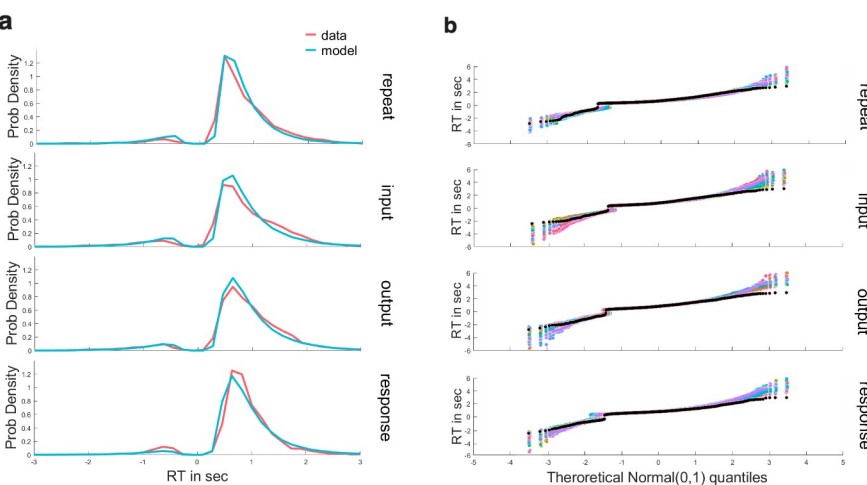

**Fig 6. Model fit.** (a) Behavioral RT distributions across the group are shown for switching at each level (red line) together with posterior predictive simulation from the HDDM (light blue) in updating trials. Distributions of correct (the right positive tail) and incorrect (left negative tail) trials show good correspondence between data and model. (b) Model fit with Quantile-Quantile plots. Model fit to behavior can be more precisely viewed using quantile-quantile plots, showing quantiles of the empirical behavioral RT distributions (black) against the 50 simulation of RT distribution (colored lines, capturing model uncertainty) from the posterior predictive of the HDDM model, for correct response (positive RT) and incorrect responses (negative RT) in updating trials. Quantiles were computed at the group level. The empirical RT was mostly within the range of the simulated RT with a small over-estimation at the right tail of the distribution.

managed simultaneously. Furthermore, in updating trials, the threshold was most strongly elevated by response gate, such that any additional conflict at the cognitive level (by input or output switch) did not further elevate the threshold much when response conflict was present. Conversely, in maintenance trials, input gate switching (i.e., switching to a maintenance policy by closing the gate) had the strongest impact on threshold whenever another gate was switching with it. We speculate about potential mechanisms for this effect in the discussion.

## Discussion

These findings provide heretofore untested empirical support for the PBWM theoretical framework in which working memory involves a hierarchy of (content-addressable) selective input, output, and response gating operations. First, we developed a task that separately taxes the need for input and output gating processes, motivated directly by the computations in PBWM. The cortico-striatal gating framework offers a theory of selective gating that includes not only plausible mechanisms for learning and generalization of gating policies [6,7,9,18,62] but also supports advanced cognitive control functions, like hierarchical control [12], higher order learning, and cognitive flexibility [11] in complex tasks like the reference-back-2 task used in this study. The previous reference-back task focused on input gating [26,27,29,32] and its correlates in PFC and BG [30]. Here, we augmented that task by adding the need to track two independent categories over trials, thereby taxing selective input gating, while also orthogonally manipulating output gating. The advantage of the reference-back-2 task is that it is a continuous task that manipulates *selective* input and output gating demands to/from a particular address in memory, while preserving the need for continued maintenance of other information across trials.

Second, we confirmed the basic prediction that neural correlates of input, output, and response gating evolve dynamically and have analogous effects on behavior. The mass-

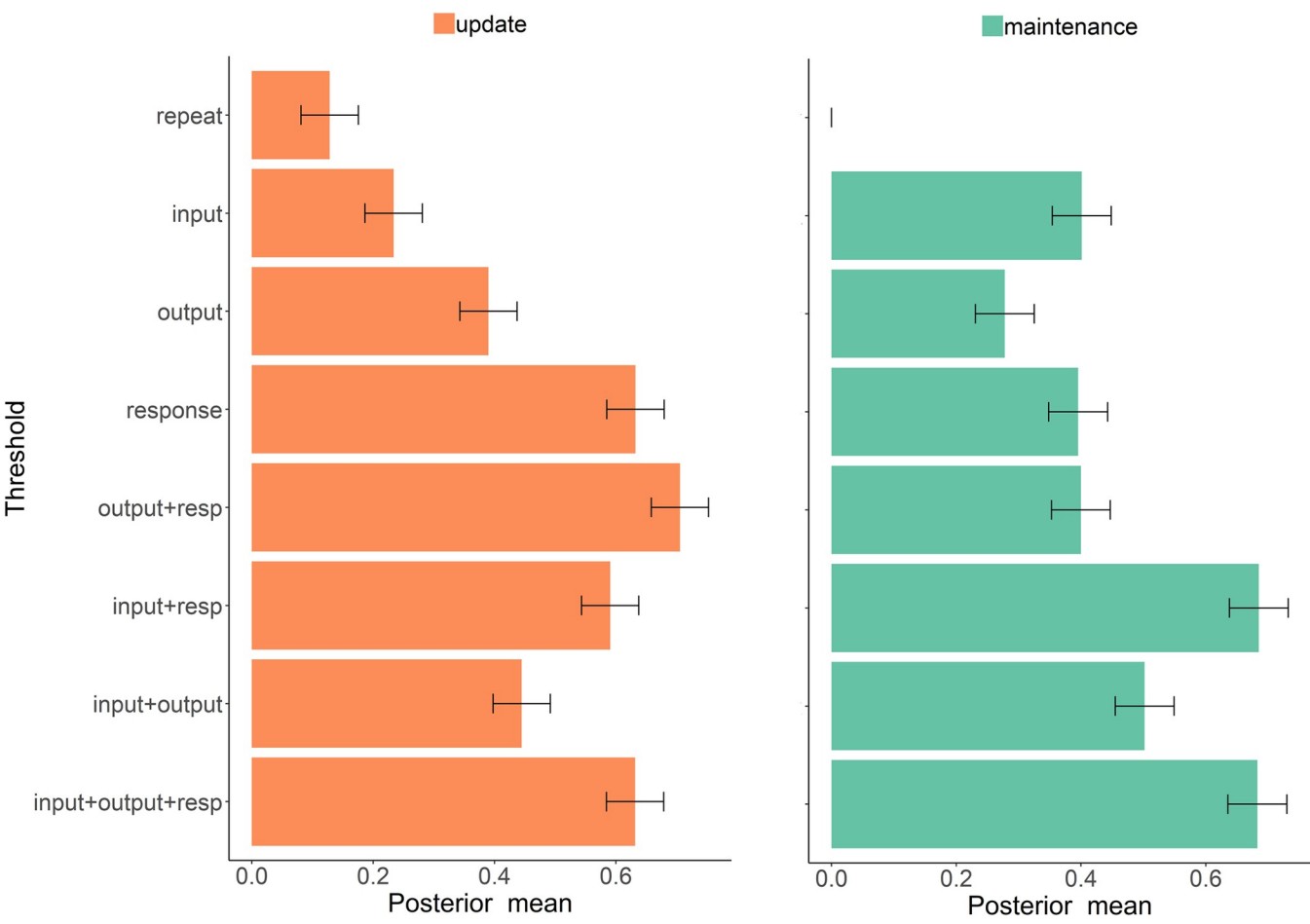

**Fig 7. Under-additive interaction between gating selections.** Decision threshold parameter estimates from HDDM in updating (orange bars) and maintenance trials (green bars), exhibit main effects of gate switching at each independent level of gating (input, output, and response), and under-additive interactions between gate switches in input-output, input-response, and output-response. Error bars reflect within-subject 95% highest density interval of the posterior distribution, in each case relative to maintenance repeat trials.

univariate analysis of the EEG provided the temporal confirmation of this prediction by showing distinct spatiotemporal patterns for each gating level (input, output, response) that were independent and partially overlapping in time (see [53] for similar conclusions in PBWM simulations and behaviorally in a hierarchical rule task). An ERP analysis of selected electrodes where the observed mass-univariate response was maximal, showed that each gating decision was marked with a different ERP component, suggesting again that each gating decision engages a separate mechanism. Specifically, input switching to an updating policy was marked by an increase in the P3b component, possibly reflecting the change in the relevance of the perceptual stimulus for the updating of WM (triggered by the switch in the task cue) [65,66]. This is consistent with previous work with the reference-back that has linked the P3b component to the categorization of the stimulus as goal-relevant [28] or as meaningful for behavioral adaptation in a context-specific manner [75]. The N2-like enhancement during output switching potentially reflects the increased need for response monitoring [67] induced by the conflict in the output gate. Such interpretation would be consistent with the predicted dependency between hierarchical gating loops such that conflict at the higher level output gate may also temporarily increase response conflict [12]. Finally, response switching was marked by an increase in the posterior PSW amplitude, possibly reflecting difficulty in selection or

preparation of an action that is different from that already planned or still in memory from the previous trial [66,76].

Third, we found that output gating (and particularly, switches therein) enhances the neural index of WM content selectively for the relevant category. The probabilistic and unpredictable switching between categories imposed a dynamic need to access distinct "addresses" within WM to update to, and read out from. This task property also had the important advantage of overcoming the challenge of measuring memory prioritization without being contaminated by the removal of irrelevant information [77] that often coincides with such prioritization manipulations (e.g., [38]). The similarity measure analysis suggested that neural representations of relevant WM categories were enhanced following WM gate switches at the corresponding levels. Analogously, neural representations of motor responses were enhanced following response switches. We thus propose that gating operations, perhaps implemented by corticostriatal circuits, are partially responsible for previous observations of prioritization. Similar conclusions have been made in attentional shift experiments in which one or another stimulus category becomes relevant, and where striatal activity dictates whether PFC selectively enhances posterior representations of the relevant category [78]. Nevertheless, further investigations are needed to determine which neural mechanisms provide prioritization gain during gate switching. For example, gate switching may change the "active neural trace" (e.g., [50]) of representations or it may increase their accessibility for read-out (e.g., in other prefrontal layers as in PBWM). Other studies suggest that prioritization does not simply amplify relevant representations but that it also modifies the formatting of the non-selected representation so that they are less similar to the form that would be read out (e.g., [79,80]). Future research should also test whether gate switching facilitates coupling between cognitive rules and relevant task features like action representations. [81,82]

Finally, the above neural and behavioral findings were further supported by quantitative computational modeling using HDDM. Hierarchical PFC-BG models suggest that conflict at the level of PFC representations recruits the same mechanism to prevent premature motor responding [12,35]. We thus assessed whether gate switches at all levels of the WM hierarchy are quantitatively related to adjustments in decision threshold that is thought to reflect cognitive control, thus related to slower but more accurate responding (Fig 7). We found that switches at any gating level were related to increases in the estimated decision threshold. This finding builds on previous studies showing that switches or response conflict gives rise to increased threshold via PFC-BG mechanisms [57–60], but extends it to confirm the hierarchical PFC-BG model prediction that such effects hold for switches at more cognitive levels [12].

Furthermore, by comparing the impacts of threshold adjustment when multiple gates are switching, the HDDM analysis provided some further insight into the observed parallel processing patterns in RT and in the EEG. First, threshold adjustments were under-additive, with largest adjustments dictated predominantly by switches in response gate for updating trials, but by switches in input gating for maintenance trials (i.e., the need to close the gate to prevent updating). We speculate that closing the gate over WM might also recruit the same prefrontal sub-thalamic pathway thought to be involved in transient response inhibition (e.g., [70,83]), therefore switching to a maintenance mode might have a larger impact on motor gating than switching to an updating mode. A limited finding that is consistent with this hypothesis is the previous observation in the reference-back task [27] that input switching to maintenance (gate closing) was marked by a mid-frontal EEG signature that has previously been associated with the triggering of the mPFC-STN network during response conflict [58,70]. Future research should test the possible similarities in the neural mechanisms that are triggered by response inhibition vs cognitive inhibition in WM.

Notably, these threshold effects were obtained even though drift rate was also allowed to vary by condition (see S4 Fig). In the DDM, (lower) drift rates can also capture slowed RT

when a condition is more difficult, as opposed to a control mechanism. Accordingly, we did observe that drift rates were slower in conditions that required more cognitive processing to succeed: updating vs. maintenance trials and switches at the cognitive level vs. switches at the response level. In both cases additional cognitive processing is required: updating trials require replacing WM content in addition to accessing previous content for responding, and switches at the cognitive levels are known to be more difficult than simple response switches (e.g., [84]). These findings boost the interpretation that gate switches at any given level are specifically related to elevated decision thresholds, while accounting for impact of difficulty on drift rate.

A key property of the cortico-striatal model that gained support by current findings is that higher order cognitive "actions" (input and output gating operations) scaffold on top of the canonical computational motor gating operations. Indeed, striatal dysfunctions such as in Parkinson's disease causes not only progressive motor degeneration but also cognitive deficits that are both related to the dysfunction of the gating system [85]. More recent work also found that degraded ability to perform selective updating of WM is the key marker of the cognitive deficits in Parkinson's [86].

Lastly, our study has various limitations. First and foremost, while the EEG method allowed us to assess the temporal dynamics of gating signals and representation decoding, it does not afford the ability to assess the involvement of corticostriatal circuits specifically. Functional imaging studies have implicated striatum in input gating in the reference back task [30], and output gating in other tasks [6], but testing the involvement of this circuit in selective content-addressable gating awaits further study.

Second, with the aim to increase our understanding of the latent cognitive processes that give rise to the full behavior in the reference-back-2 task and not just the mean RT and error rate, we leveraged the HDDM framework. However, it is likely that the decision process engaged during working memory gating diverges from that assumed by the standard DDM, and other models should be considered in future work (including those with time-varying drift rates and/or boundaries). Such models are difficult to estimate, but recent tools open the door for such investigations [87]. Moreover, future research could also employ a model-based approach where neural patterns are linked to cognitive mechanisms through computational models (e.g., [58,88]).

## Methods

### Ethics statement

All participants were compensated for their participation and gave written informed consent as approved by the Human Research Protection Office of Brown University under protocol 0901992629, "Learning and Decision Making Genetics and EEG"

### Participants

Thirty-two right-handed adults (aged 18–24; 18 female) with normal or corrected-to-normal vision completed the experiment. All spoke English natively, were screened for neurological medications/conditions. Two participants were excluded from the analysis, due to technical problems with the experiment. All behavioral and EEG data has been made available on Dryad (https://doi.org/10.5061/dryad.00000002t [89]).

### Stimuli and procedure

Stimuli presentation and behavioral data collection were implemented using the Psychophysics Toolbox extensions in Matlab [90–91]. One out of four possible stimuli was presented on

each trial. The stimuli were each of a distinct category: letters ("X", "O") and symbols ("%", "#") with two stimuli in each category. The stimuli appeared in a random order. Each trial started with a presentation of a stimulus inside a colored frame (red or blue), that indicated whether it was an updating trial or a maintenance trial. Participants had to make a matching decision between the presented stimulus and the last item category that was presented inside the updating color (e.g., red). Therefore, letters had to be compared with letters, and symbols with symbols. For all trials (regardless of blue or red) required participants to follow the same rule of comparing the stimulus with that of the corresponding category in the most recent updating trial. Participants were instructed about the meaning of the color (i.e., which one corresponded to reference to be updated and which one indicated the stimulus should only be compared to that in memory). The selected reference color (red or blue) was counterbalanced between participants. The color of the frame in each trial was biased with 75% probability for one color in the first six blocks of the experiment and 75% probability for the other color for the last six blocks of the experiment. The order of color bias was chosen randomly. The biased color manipulation allowed us to have a more stable measure of selective output gating where the relevant category changed (output switch) but fewer trials in which there was also a color (input policy) switch. Participants were instructed that there will be a color bias that will flip once during the experiment. "Same" and "different" responses were indicated by using the right and left index fingers, respectively, to press 'Z' and '/' on the keyboard. Response mappings were chosen randomly for each participant.

Stimulus presentation was limited to 3 sec. The response was followed by an inter-trial interval that was jittered between 800–1000 ms. Participants were instructed to keep their eyes fixated on the center of the screen throughout the experimental blocks. The first two trials in a block were always updating trials with stimuli from the two categories. Participants were instructed that accuracy will not be measured in the first two trials in the block. The experiment comprised of 12 blocks, including 90 trials each. Participants had to reach 80% accuracy on the practice block before they began the experiment. Participants were allowed to repeat the practice block up until 4 times.

## Electroencephalogram (EEG) Recording and processing

Scalp voltage was measured using 62 Ag/AgCl electrodes referenced to a site immediately posterior to Cz using a Synamps2 system (bandpass filter 0.5–100 Hz, 500-Hz sampling rate). Pre-processing was conducted using the EEGLAB and ERPLAB toolboxes [92–93]. During pre-processing, data were low-pass filtered at 30 Hz and high-pass filtered at 0.1 Hz. Epochs were segmented from -200 to +800 ms surrounding stimulus onset and were baseline corrected from –200 to 0 ms before the onset of the stimulus. The epoched data were visually inspected and those containing large artifacts due to facial electromyographic (EMG) activity or other artifacts (except for eye blinks) were manually removed. Independent components analysis (ICA) was next conducted using EEGLAB's runica algorithm. Components containing blink, oculomotor artifacts, or other artifacts that could be clearly distinguished from genuine neural activity signals, were subtracted from the data.

## Data processing for univariate EEG analysis

To extract the neural correlates in the EEG signal of conditions of interest we employed a mass univariate approach. A multiple regression analysis was conducted for each participant, in which the EEG amplitude at each electrode site and time point was predicted by the conditions of interest while controlling for other factors such as RT (such an approach was recently used [63,64]). For the regression analysis, the EEG signal recorded with 500 Hz sampling rate was

down-sampled by a factor of 4, resulting in 125 time points for the selected window of -200:800 ms around stimulus onset. The EEG signal was z-scored before it was entered to the robust multilinear regression analysis to account for remaining noise in the data [64].

Two separate multilinear regressions were run. The first regression assessed the neural correlates for gate switching. It included 7 regression factors: log of RT (to remove variability due to slower responses in some conditions; [64]), and 6 contrasts dummy coded as 1 and 0: WM state (updating vs maintenance), input gating (WM state switch vs repeat), output gating (category switch vs repeat), response gating (action switch vs repeat), bias (frequent updating vs rare updating), and finally one interaction between WM state × input gating (to search for a selective input gating pattern in updating trials).

The other regression assessed the neural prioritization of representations, i.e. to identify an EEG signal that differentiates between representations at each gating level. The second regression included 4 factors: log of RT and 3 contrasts dummy coded as 1 and 0: WM state (updating vs maintenance), category (symbols vs letters) and action (same vs different).

## Statistical analysis of GLM weights

Statistics on the regression weights were performed across participants for all electrodes and time points by testing the significance of each point against 0. To correct for multiple comparisons, we performed cluster-mass correction by permutation testing [94] with custom written Matlab scripts. All analysis code has been made available on Zenodo (https://doi.org/10.5281/zenodo.4623800 [89]). Cluster-based test statistics were calculated by taking the sum of the t-values within a cluster of significant points with threshold for a t test significance level of P = 0.001. This was repeated 1000 times, generating a distribution of maximum cluster-mass statistics under the null hypothesis. Only clusters with greater mass than the maximum cluster mass obtained with 95% chance permutations were considered significant [64]. The results of the second regression analysis yielded significant univariate patterns for the two representation types (category and action). Note that a different regression model was also run to assess the neural correlates of stimulus representations. This regression produced a significant univariate pattern only for the "O" representation but not for any of the other stimuli. Therefore, we did not continue with further analyses related to stimulus similarity.

## Trial-by-trial similarity index

Using the GLM masks we then computed the dot product between individual trials (voltage maps of electrode * time) and the identified masks (electrode * time maps of t-values of significant pixels; [64]). This computation produced a trial-level similarity measure (see Fig 5A) that presumably reflects how similar the EEG signal of the probed representation in memory on a given trial to the mask activity of this representation. To visualize the temporal dynamics of the EEG index (Fig 5B), the similarity measure procedure was calculated in sliding time bins of 40 ms across the epoch, and applied only to time bins in which the original mask was significant. The similarity index was calculated 121 times in each trial, with a 32ms overlap between time bins. The temporal similarity indexing was obtained by calculating the mean similarity between trials where one representation was probed and when the other one was probed. The effect of gate switch and repeat on the similarity index was calculated using t-tests on the individual time bins.

Note that to test the effect of gate switching on prioritization, we excluded trials from the temporal dynamic analysis where the same stimulus was repeated from the previous trial. This exclusion allowed us to control for impact of perceptual switches that could have elicited an involuntary stimulus-driven enhancement of memory representations that match the probed

category stimulus (e.g., [28,95]). Therefore, we restricted analysis to output repeat trials involving a perceptual change (e.g., from "X" to "O"), while output switch trials also included a category change (e.g., from "X" to "%").

## Drift diffusion modeling

The DDM is a common sequential sampling model of two-choice RT tasks. The advantage of this model is that it can translate response time distributions and error rates to the underlying generative parameters in each task condition, and it has previously been used to summarize decision dynamics that arise from BG gating [35,58–60]. The core parameters in the model are threshold (or boundary separation) $a$, drift rate $v$, and non-decision time $t$. Threshold is the distance between the response boundaries, where higher threshold indicates that more evidence needs to be accumulated before committing to a choice, leading to slower but more accurate responses. Conflict or switching in motor responses leads to elevated decision thresholds [35,58–60]. Drift rate is the rate in which evidence is accumulated. Larger drift rates are usually interpreted to reflect higher quality of evidence that is expressed by faster and more accurate response times. Finally, non-decision time captures the processes that are not related to the decision making process such as stimulus encoding and motor execution.

We tested whether the DDM provides a good model of RT distributions of correct and incorrect choices during the reference-back-2 task in which the decision threshold can be adjusted as a function of conflict or switches at any gate level (while allowing the drift rate to also vary by condition)). The underlying decision-making process during the reference-back-2 task were estimated from the DDM likelihood functions [35]. We used hierarchical Bayesian estimation of DDM parameters, where individual's fit is constrained and informed by the group distribution leading to more accurate estimation of parameters at both the individual and group level [35].

Parameter estimation in the Hierarchical Bayesian framework used the Markov-chain Monte-Carlo (MCMC). HDDM is especially beneficial for estimating individual parameters while optimizing the tradeoff between random and fixed-effects [35]. The first two trials in each block, omission trials, and trials with very fast RT ($<$200ms) were excluded from the analysis. RT was limited to 3sec. There were three thousand samples generated from the posterior using four chains. The first thousand (burn-in) and every second (thinning) were discarded. Proper chain convergence was tested between the MCMC chains, using the Rˆ statistic [96], which measures the degree of variation between chains relative to the variation within chains. The maximum Rˆ value across all parameters in all eight models was 1.03, indicating that all chains converged successfully [97].

Statistical analysis was performed on the group mean posteriors. The Deviance Information Criterion (DIC) was used for model comparison [98] which balances model fit against complexity. We first tested the simplest model where the five conditions (WM state, input switching, output switching, response switching, and updating frequency) were used as regression weights without interactions and with group level estimates (DIC: 29339). We systematically added interactions, and the model with the best fit was one in which drift rate and threshold exhibited 4-way interactions (DIC 28806). This best fitting model was estimated again allowing for subject-level estimates in each condition (DIC 27232):

Threshold ~ $a\_mr$ + Update_frequency + (WM state × input × output × response) +. . .
Drift rate ~ $v\_mr$ + Update_frequency + (WM state × input × output × response).

$A\_mr$ is the intercept for threshold and $v\_mr$ is the intercept for drift-rate, Update_frequency reflects the high or low frequency of updating, and the 4-way interaction captured all combinations of switch or repeat during updating and maintenance trials (15 combinations

in total). Within subject regressions were used such that all gate switch effects are evaluated relative to the maintenance repeat condition (the intercept), using patsy in HDDM. Significance was determined if the 95% confidence interval of the posterior mean did not overlap with 0.

## Behavior characterization and model validation

We first plotted the overall RT distributions for each gating condition and interaction, separately for updating and for maintenance trials (Figs 6 and S2) using Gramm plotting toolbox [99]. This plot showed that the HDDM model captures choice proportions (of correct and incorrect trials) and the different shape of the RT distributions within each gating condition, simultaneously. A critical test of a the DDM model is that it can capture the full RT distributions in each condition. Although we used model fit statistics (DIC) to select the best-fitting model, it is also important to validate that the best fitting model can capture the observed response proportions and RT distributions. For this model validation we generated quantile-quantile plots describing the correspondence between behavior and DDM predictions (Figs 7 and S3) [99]. We compared the shape of the empirical RT distribution to the shape of the simulated RT distribution by plotting each data against a theoretical normal distribution. The plots in Figs 7 and S3 exhibit the RTs on the y-axis as a function of the estimated quantiles on the x-axis based on the inverse of the continuous cumulative distribution function (quantile function) that was estimated at the group level. RTs of the model were simulated 50 times from the posterior predictive of the HDDM model and plotted on top of each other to show the uncertainty in the model. As can be seen, the empirical RT was mostly within the range of the simulated RT with a small over-estimation at the right tail of the distribution.

## Supporting information

**S1 Fig. Behavior effects of gate switching.** Mean RT (A-C) and Error rate (D-F) demonstrate differences in performance across all levels of gate switching and interactions between gating levels (output and response, input and response and, input and output) in maintenance trials (left panels) and updating trials (right panels).
(TIF)

**S2 Fig. Model fit.** Behavioral RT distributions across the group are shown for switching at each level (red line) together with posterior predictive simulation from the HDDM (light blue). Distributions of correct (the right positive tail) and incorrect (left negative tail) trials in updating trials (left) and maintenance trials (right) show good correspondence between data and model.
(TIF)

**S3 Fig. Model fit with Quantile-Quantile plots.** Model fit to behavior can be more precisely viewed using quantile-quantile plots, showing quantiles in updating (left) and maintenance (right) trials, of the empirical behavioral RT distributions (black) against the 50 simulation of RT distribution (colored lines, capturing model uncertainty) from the posterior predictive of the HDDM model, for correct response (positive RT) and incorrect responses (negative RT). Quantiles were computed at the group level.
(TIF)

**S4 Fig. Drift rate parameter estimations from the HDDM.** Drift rate in updating (orange bars) and maintenance trials (green bars) exhibit slowing (lower drift) in conditions that required more cognitive processing and exhibit facilitative interactions between gate switches. A possible interpretation for the facilitation finding is that WM updating decisions increase

the mutually facilitative effect of switching across the gating system. The relative drift rates are presented as positive for plotting convenience.
(TIF)

**S1 Text. The extended behavior results.**
(DOCX)

# Acknowledgments

We thank Romy Frömer and Matt Nassar for providing code used in the EEG analyses.

# Author Contributions

**Conceptualization:** Rachel Rac-Lubashevsky, Michael J. Frank.

**Data curation:** Rachel Rac-Lubashevsky.

**Formal analysis:** Rachel Rac-Lubashevsky.

**Funding acquisition:** Rachel Rac-Lubashevsky, Michael J. Frank.

**Methodology:** Michael J. Frank.

**Supervision:** Michael J. Frank.

**Writing – original draft:** Rachel Rac-Lubashevsky.

**Writing – review & editing:** Michael J. Frank.

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
