## [Decision Letter · Decision Letter 0]

18 Feb 2021

Dear Dr. Rac-Lubashevsky,

Thank you very much for submitting your manuscript "Analogous computations in working memory input, output and motor gating: Electrophysiological and computational modeling evidence" for consideration at PLOS Computational Biology.

As with all papers reviewed by the journal, your manuscript was reviewed by members of the editorial board and by several independent reviewers. In light of the reviews (below this email), we would like to invite the resubmission of a significantly-revised version that takes into account the reviewers' comments. In particular, the authors should endeavour to improve the readability of the manuscript and clarify the presentation of their figures and results. 

We cannot make any decision about publication until we have seen the revised manuscript and your response to the reviewers' comments. Your revised manuscript is also likely to be sent to reviewers for further evaluation.

Sincerely,

Daniel Bush

Associate Editor

PLOS Computational Biology

Samuel Gershman

Deputy Editor

PLOS Computational Biology

Reviewer's Responses to Questions

**Comments to the Authors:**

Reviewer #1: “Analogous computations in working memory input, output and motor gating: Electrophysiological and computational modeling evidence,” Rac-Lubashevsky and Frank. This is an interesting and carefully carried out EEG study of the “reference-back-2” working memory task that uses quantitative fits to the PBWM model to derive EEG correlates of gating operations specified by PBWM. This work makes a potentially very impactful contribution to the study of the neural bases of cognitive control because, as the authors state in the Introduction, “to date the gating WM literature and the WM prioritization literatures have not been linked.” Indeed, this statement could have been lifted from a grant proposal submitted by this reviewer just two months ago. My overall assessment is that I’m very enthusiastic about this work. However, there are several points where the exposition could be clearer, and some where the authors might consider making contact with other literature.

Overall, the paper would be easier to read if the Introduction were more coherent with the rest of the manuscript. For example, there are several constructs and/or distinctive empirical patterns that pop up in the Results section that relate to ideas introduced in the Introduction, but for which different words are used. For example, in the Intro we read that “switches give rise to a transient period of conflict;” but only in the Results are we told why: “In PBWM, gating signals are used to displace prefrontal activity states with new information.” This same section of the results introduces the importance of under-additive interactions, and these are illustrated in Figure 3. Only further along in this section is it explicitly stated that hierarchical gating processes that “evolve partly in parallel lead to under-additive interactions.” This is an important pattern in the data that returns several times, and its significance really should have been introduced in the portion of the Introduction that introduces the architecture of the PBWM. Additionally, the at-face-value contradiction of a hierarchically organized set of operations and “partly” parallel engagement needs to be explained more explicitly. E.g., even later we will encounter HDDM modeling of gating operations; is it perhaps possible that there is leakage from the decision that’s underway at, say, the input gating level that gets the output gating operation started before input gating is finished? Whether or not this is true, there are presumably quantifiable dynamics in PBWM that would provide a more rigorous level of understanding than just the verbal statement that some of these processes can be “partly” parallel. (Additionally, whatever the answer is, it’d be helpful if an illustration of these dynamics could also be incorporated into Figure 1; perhaps a timeline of when different elements of the model are engaged?)

It strikes me as backward to say that “WM prioritization is thought to support ‘controlled attention’ by enhanding top-down influence…” Isn’t WM prioritization itself an example of controlled attention?

Is “BG gating” (p. 9) different from “gating”?

Contact with other literature. I’m hesitant to write this, because the authors already do a good job of citing a lot of the relevant literature, but there are some additional areas where contact can be made with work that one doesn’t necessarily classify as “PLoS Computational Biology”. The first is that the reference-back-2 is reminiscent of the “Garavan task,” a running span task that Hugh Garavan published back in the 90s(?) that, for a while at least, captured a lot of interest in the cognitive psychology of cognitive control. A second oldie is Marcia Johnson’s MEM, particularly the ‘refresh’ operation in relation to the increased decodability described here. Indeed, one of the things that so appealing about models like PBWM is that they posit explicit quantifiable mechanisms that can either ‘confirm’ and elaborate on older verbal models (as may be the case here with ‘refresh’), or, in other cases, provide better accounts. More recently than MEM, studies of retrocuing effects in working memory, some of them already cited, are germane. One question I’m thinking about came up in a finding of ‘enhanced decodability’ that Sprague and Serences interpreted as evidence for reactivation of ‘activity-silent’ components of a stimulus representation, but that others argued might just be akin to the sharpening one can see in bump attractor models when more energy is added. Do the present findings shed light on this, or is there perhaps a ‘future direction’ with this approach that could do so?

Finally, thinking about Figure 4, is there a way to relate any of the “gating components” extracted by the model to more classical ERP components, or perhaps to something like frontal midline theta?

Other comments

- There’s reference to RL in the intro, but no apparent explicit role for RL in the Results. But presumably there are prediction errors associated with many of the operations carried out in performing this task. Even if they’re not explicitly modeled here, some consideration of how they would be incorporated into a more complete simulation would be useful? This is also another chance to make contact with other literature, in that the model that underlies the Olivers and Roelfsema (2020) commentary (which is cited) very explicitly incorporates principles of RL on a trial-by-trial basis.

- The comparisons of 2-back vs. reference-back-2 tasks feel forced. For example, on one hand, reference-back-2 could also be thought to require “encoding, inhibition, binding, matching, maintenance, updating and removal,” and on the other hand, these operations derive from intuition, but people have successfully trained 3-layer RNNs to perform n-back tasks, and these models don’t explicitly implement most of these operations.

- Relatedly, in the Discussion (page 22, Line 485), the authors claim that the reference-back-2 task can assess prioritization without being contaminated by removal. However, when a certain category is prioritized (say, letter) and the item needs to be updated (X -> O), the old item still needs to be removed.

- Figure colors: The authors tend to use very similar colors in figures which are hard to distinguish. For example, Figure 1 uses light and dark orange, and bar graphs in later figures were colored with slightly different shades of green, blue or orange. Maybe consider using colors with higher contrast.

- Figures 5 and 8 have a lot of bar graphs that are difficult to grok.

a. All the figures look kind of similar and it is not easy to tell what metrics and what kind of comparisons they are showing without perusing the figure captions and main text. The authors can consider using letter indices (A, B ..) instead of “left corner”, “top” to refer to specific graphs. Consider using titles and sub-titles for columns/rows to refer to groups of figures, e.g., for the 4 RT graphs and the 4 Error Rate graphs.

b. Even though in each group there are 4 figures with 4 bars each, upon a closer look, there are only 8 distinct bars (repeat, input, output, resp, input+output, input+resp, output+resp, and all-sw), with a lot of redundancy. I understand the authors want to showcase the separate interactions, but maybe they can condense the plots into 1 or 2, or think of a clearer and more concise way to present the data.

c. Consider using asterisks and brackets to indicate significant main effects and interactions: right now it’s hard to see what the authors are trying to show us. Also it is not easy to locate the “under-additive interactions” just by looking at the bars. Highlight them in some way.

- Figure 5 (page 16):

a. Panel a is not clear. It seems to be a cartoon/schematic as we don’t know what the GLM mask is for (category or response). Moreover, the text around the heatmap is too small to decipher, the figure titles are not informative, and the axes are not labeled. And, where is the “red rectangle” that the caption referred to Line 369)?

b. In Panel b, it might be better if authors can have a significance bar that indicate time frames with significant differences between switch and repeat conditions. And why the big swaths of white on the bottom row?

- Both Figures 6 and 7 show that the HDDM fits the data well. Maybe one figure is sufficient and move the other one to the Supplemental Materials?

- Page 20: Line 440 says “drift rate was also allowed to vary by condition” whereas Line 449 says that the decision threshold effect holds “while accounting for impact of difficulty on drift rate.” I thought “accounting for …” means to “hold constant”? Please clarify

Why is drift rate negative? Does this correspond to a reduction of drift rate? Because drift rate also increases as it goes from high to low (response) level, why is the story mainly about the decision threshold but not about drift rate?

- Linking gating and prioritization: seems like looking at the similarity of EEG pattern to the masks that care about the representation (category or response) is a weird way of decoding… The word “decoding” gets used a lot, and often not very precisely. Here, you are not decoding an item’s identity per se, but rather looking at how much the brain signal looks like the pattern that is distinguishing the items?

- P. 23, line 509: should it be “than” instead of “then”?

Reviewer #2: PCOMPBIOL-D-21-00030

Analogous computations in working memory input, output and motor gating:

Electrophysiological and computational modeling evidence

By Rac-Lubashevsky and Frank

SUMMARY

This study utilized behavior paradigms, EEG, decoding analyses of EEG signals, and a DDM to evaluate the PBWM (prefrontal cortex basal ganglia working memory) model that posits working memory involves a hierarchy of selective input, output, and response gating operations. They adapted the reference-back task, which had previously been used to study PFC-BG connections related to input gating, to require subjects to track two independent categories over trials, which allowed the experimenters to require input gating while orthogonally manipulating output gating. Univariate analysis of EEG data showed a confirmation of their hypothesis that neural correlates of different gating types evolve dynamically and have similar behavioral effects. They found that motor responses were enhanced following responses switches, and that the decodability of the relevant WM category was enhanced after an ‘output gate switch’. This could result from a change in the “active neural trace” or an increase in the accessibility for “read-out”. Regardless, they argue that the cognitive gating operations are implemented by similar cortico-striatal circuits that have been linked to motor gating operations. The authors then used computational modeling, HDDM (hierarchical Bayesian estimation of DDM parameters), to corroborate their findings. They found that switches at any gating level related to increases in the decision threshold. Finally, the researchers also described the “under-additive” interactions when more than one type of switching was needed. The results suggest that actions selection is at least partially parallel, and that decisions are processed even while the identity of the cognitive rule is uncertain.

EVALUATION

This study is well motivated and addresses and important question about the neural mechanisms of working memory. The reference-back task paradigm is an appropriate choice and well suited for the EEG approach taken. The addition of the DDM modeling allowed them to support and extend their findings. My biggest area of concern is in how the data are presented and how the results are explained (e.g., in Figs 3, 6, 7, and 8). The plots are too cluttered, with insufficiently contrasting colors for different conditions, and without any visual markers of statistical significance for key findings. Also, while the DDM modeling was used to support their findings, its inclusion is not sufficiently justified and should be more clearly motivated.

MAJOR CONCERNS

1. THEORY EXPLANATION. The description of the PBWM architecture in Figure 1 is quite confusing. There are so many different uses of colored arrows, colored arrow heads, colored letters, yellow squares, etc. It is really challenging to follow and does not serve the authors well in explaining the proposed gating model. This figure should be reconceptualized and more clearly explained to the reader.

2. DATA PRESENTATION. The presentation of the data severely limits their interpretability and impact. For example, Figs 3, 6, 7, and 8 present large piles of data that are insufficiently labeled, poorly differentiated by subtle color variations (dark green vs. light green), and poorly organized so that interpreting the results is challenging if not nearly impossible. I cannot emphasize how much the use of “dark green / light green” contrasting colors makes these plots even worse. This comes up often and cannot be overlooked. Many subplots in these figures lack titles or other organizational signposts to help explain the results. The Results section is littered with statistical comparisons and significance results which are not found in the figures referenced. This puts the onus on the reader to “figure things out” which is not acceptable. Also, bar plots are presented without the underling individual data points, and often it is not clear if the error bars shown are STD, SEM, or 95% CI.

3. MODEL JUSTIFICATION. The use of the computational model is not sufficiently justified in the paper. It is unclear what value is added by the modeling that was not already found from the EEG results. I believe their *is* value added, but this needs to be spelled out in detail for the reader. In the Introduction, the authors say that the DDM can approximate BG dynamics, which is helpful, but not sufficient motivation for the added complexity (and inherent assumptions) of the model. The indecipherable Figure 8 compounds this issue by obscuring the results of the model. The metrics in this figure are explained in a single line: “We thus assessed whether gate switches at all levels of the WM hierarchy are quantitatively related to adjustments in decision threshold (thought to reflect cognitive control, thus related to slower but more accurate responding), or whether any such effects could be attributed to alterations in drift rate (thought to reflect increased difficulty and thus slower and less accurate) (Fig. 8).” This needs to be fleshed out more, especially for readers who are less familiar with DDMs and how their parameters may correspond to different neural mechanisms. Finally, near the end of the Discussion, the authors note a limitation of the HDDM framework that opens up more questions: “other models should be considered in future work (including those with time-varying drift rates and/or boundaries)”. It is unclear why their current model cannot address these issues.

4. WHY THE BG? The only evidence presented that supports the supposition that the BG is involved in the gating mechanisms being targeted here is based on the HDDM model, which itself is built on assumptions. There is no direct evidence in the EEG data of the BG involvement, of course, because of the ‘inverse problem’ and the impossibility of localizing the source of EEG signals. However, there are analytic approximations (e.g., Loreta variants) for localizing electrical potentials from scalp recordings. Some approach like this should be used attempted to justify the neural circuit claims of the PBWM model. Relying on assumptions of the HDDM model alone to describe the involvement of the BG is insufficient without a significant amount of further explanation and justification.

MINOR CONCERNS

a. Fig. 5 notes that it contains a “red rectangle” to indicate the sliding window for the similarity index, which is not present. That section of the figure, panel a, is confusing, with impossibly small labels, and overall hard to interpret.

b. The concept of “under-additive” interaction should be better explained in the text. This is a concept that may not be self-evident and as a reader I had to spend time try to understand the specific point made in the context of this paper.

c. Figs. 6 and 7 could easily be combined into one smaller plot. The display of every level in multiple subplots is unnecessary when the main take away is a lack of difference.

d. Fig. 5b is using SEM, not 95% Cis, for error shades, so significance between conditions needs to explicitly marked in the figure, else the errors should be switched to 95% CIs.

e. Fig. 1 could be improved by following conventions from other model diagrams, such as an LSTM unit: https://www.researchgate.net/figure/The-structure-of-the-Long-Short-Term-Memory-LSTM-neural-network-Reproduced-from-Yan_fig8_334268507.

**Have all data underlying the figures and results presented in the manuscript been provided?**

Reviewer #1: Yes

Reviewer #2: Yes

PLOS authors have the option to publish the peer review history of their article (what does this mean?). If published, this will include your full peer review and any attached files.

Reviewer #1: No

Reviewer #2: No
---

## [Decision Letter · Decision Letter 1]

17 Apr 2021

Dear Dr. Rac-Lubashevsky,

We are pleased to inform you that your manuscript 'Analogous computations in working memory input, output and motor gating: Electrophysiological and computational modeling evidence' has been provisionally accepted for publication in PLOS Computational Biology.

Best regards,

Daniel Bush

Associate Editor

PLOS Computational Biology

Samuel Gershman

Deputy Editor

PLOS Computational Biology

Reviewer's Responses to Questions

**Comments to the Authors:**

Reviewer #1: It is interesting and important, but kind of "inside baseball" for the general public.

Reviewer #2: The authors have sufficiently addressed our concerns.

**Have all data underlying the figures and results presented in the manuscript been provided?**

Reviewer #1: Yes

PLOS authors have the option to publish the peer review history of their article (what does this mean?). If published, this will include your full peer review and any attached files.

Reviewer #1: **Yes: **Brad Postle

Reviewer #2: No

**Have the authors made all data and (if applicable) computational code underlying the findings in their manuscript fully available?**

Reviewer #2: Yes

---

## [Editor Report · Acceptance letter]

2 Jun 2021

PCOMPBIOL-D-21-00030R1 

Analogous computations in working memory input, output and motor gating: Electrophysiological and computational modeling evidence

Dear Dr Rac-Lubashevsky,

I am pleased to inform you that your manuscript has been formally accepted for publication in PLOS Computational Biology. Your manuscript is now with our production department and you will be notified of the publication date in due course.

With kind regards,

Zita Barta
